# Cash-Flow Schedules Optimization within Life Cycle Costing (LCC)

**Jerzy Rosłon** [1], **Mariola Książek-Nowak** [1], **Paweł Nowak** [1] and **Jacek Zawistowski** [2,*]

1   Civil Engineering Faculty, Warsaw University of Technology, 00-637 Warsaw, Poland;
    j.roslon@il.pw.edu.pl (J.R.); mariola.ksiazek@il.pw.edu.pl (M.K.-N.); p.nowak@il.pw.edu.pl (P.N.)
2   Polish Association of Building Managers, 00-688 Warsaw, Poland
*   Correspondence: jacek.j.zawistowski@gmail.com; Tel.: +48-22-234-6515

**Abstract:** Investment and construction plans, architectural and construction decisions, and spatial and technology-related decisions made at the early stages of a project have a significant impact on meeting the investment goals and customer expectations. Decision making is a very time-consuming and complicated process (due to the complexity of construction processes). The whole difficulty comes to specifying the appropriate criteria for assessing the given activities, providing answers to the questions of the decision-making bodies. A set of appropriate criteria and mathematical tools (such as computer algorithms with multi-criteria analysis) can significantly improve and accelerate the decision-making process. This article combines ESORD (an IT tool that allows you to compare different types of solutions based on mathematical calculations) with the Monte Carlo method. The developed approach can help the investor to optimize their cash-flow schedule. The original method enables the client to select a construction project variant characterized by the best economical and sustainable parameters, while taking into account customers' demands.

**Keywords:** cash flow; construction project; multi-criteria decision-making; optimization; investment variants; sustainability; LCC—Life Cycle Costing

## 1. Introduction

One of the most difficult problems faced by project managers/investors is the selection of the best variant of the implementation of a given investment plan. The struggles related to this issue appear already at the investment planning stage, after the expectations and requirements of the investor/client are defined in the functional and operational program. At individual stages of the project life cycle, the analyzed phenomena are very complex, which results mainly from the specific requirements, the complexity of the issues discussed and the nature of the technological and construction processes and the relationship between them [1]. The choices made have a significant impact on the cash-flow schedule of the investment. By cash-flow schedules authors understand timing of different project incomes and outcomes, sometimes with delays and/or acceleration (sometimes also unfavorable) [2]. From an economic perspective, the problem is especially important during the first stages of a construction project. That is because the sooner the decisions are made, the more impact they will have on the life cycle cost and key performance indices of the project [3].

Cash-flow schedules optimization is balancing the required issues, including capital structure, working capital policies, pricing, vendor costs, projects portfolio, etc., all based on the properly prepared operational workflow [2,3].

Decision making is an integral part of any field of art and science. A decision chain is a set of activities that result in the selection of a specific set of activities. Directly involved in the decision-making process is the decision-maker, who expresses specific preferences, assesses the possibilities, forecasts

the results, and selects the final variant of further actions [1]. The analysis of the current investment situation is the first challenge that the assessor has to face. A decision-making situation is a set of all elements independent or dependent on the decision-maker that directly contribute to the final result of the intended actions. In the process of formulating a given problem, factors independent of the decision-maker include a set of the so-called conditions limiting the decision (tested and possible variants), while factors dependent on the evaluator include other criteria for evaluating solutions, which are described by the economic and technical indicators most adequate to the decision-making situation, expressed in individual units [1]. The decision on further actions is difficult not only due to the complexity and marking of possibilities, and the degree of complexity of the task, but also due to the client's expectations. On the other hand, the preferences of the decision-maker largely depend on the point of view of the expert who generated the given assessment or opinion. The authors of this article, having in mind the above description, argue that the computer implementation of the computational algorithms of selected decision-making methods is an effective tool that improves the decision-making process, while obtaining ordered output data. Authors decided to combine multi-criteria decision-making methods with computer simulations in order to provide the investor with the tool for cash-flow schedule optimization. Such a tool will help the client to select a construction project variant characterized by the best economical and sustainable parameters. The method is described in detail in Section 2.3. The results with a case study are presented in Chapter 3. The method takes into account the stochastic character of predicted sales. This aspect of the solution is especially important in times of uncertainty, such as during the current COVID-19 pandemic, as, for example, it helps investors and clients to keep social distance [4].

## 2. Materials and Methods

### 2.1. ESORD IT tool

The ESORD IT tool was developed by the authors' team for optimization of decision making in construction. The principle of ESORD operation is based on the so-called client-server architecture, enabling mutual communication between the client (e.g., user's computer) and the server (e.g., database systems). The above solution allows the user to access the system from any place (computer) connected to the internet. Installation of the ESORD program is not necessary and does not require the installation of additional software, because ESORD is a website available at http://esord.pl (after login). The client of the system is a user with a computer with any web browser (for example Mozilla Firefox or Internet Explorer) with enabled support for JavaScript and Flash objects. The server part of the ESORD program is based on the following applications:

- lighttpd web server (serving as an application server, interface between the user and the entire system),
- PHP programming language (supporting system logic),
- Smarty templates (supporting the graphic layer),
- MySQL database,
- memcached (a cache system that allows you to write data).

The ESORD program distinguishes (recognizes) two types of users, that is:

- Type I users, the so-called "searching ones": this group is interested in finding a suitable variant for them (e.g., a building object, a flat). They communicate (define) their preferences to the program by completing the Residential Building Assessment Survey. Based on the completed questionnaire, the program searches for variants using the implemented algorithms of selected multi-criteria evaluation methods (for example, ELECTRE, ideal point, entropy). In the final stage, the results of the performed calculations are generated. Type I users cannot modify internal program structures.

- Type II users, the so-called "variants creators" (e.g., investments): this group can be identified as representatives of developers who add investment variants to the program. A variant is added by filling in the form available under the "Add variant" tab. After completing this form, the program asks you to fill in the questionnaire concerning this variant; it is the same questionnaire that is filled in by those who are looking for the solution (searching ones), but it does not define expectations. It describes in detail the condition of a given investment. The group of users with these rights has been defined by the program administrator. See screenshot of ESORD at Figure 1.

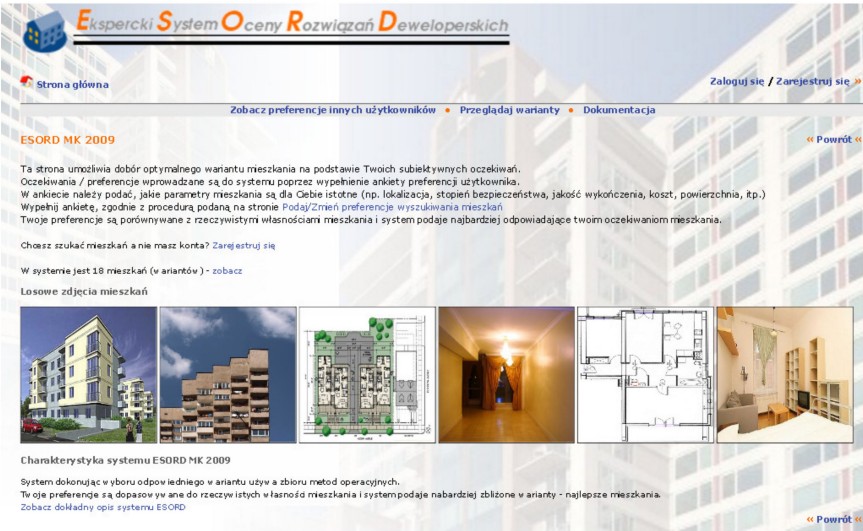

**Figure 1.** Main screen of the ESORD IT tool [own source].

The subjective preferences specified by the Type I user are compared with the actual properties of the analyzed variants. The ESORD program presents the variants that best meet user expectations. Based on the results of the multi-criteria assessment, the best variant is generated within the framework of the computational algorithms used by the system. Before the actual mathematical calculations are performed, the variants are pre-searched (filtered) on the basis of user-defined general selection criteria (for example, location—the city). ESORD prompts the Type I user to select the calculation methods to be used in the calculation process (such as AHP or fuzzy logic). The user can view the results for individual variants, achieved by different calculation methods, and can also aggregate final results. The calculation methods implemented in the program take, as input data, the vector of weights of the main criteria and the table of standardized grades. The vector of main criteria weights contains the normalized values of user preference scores with respect to the main criteria.

The ESORD IT tool (Expert System of Developer Solutions Assessment) includes kinds, types and groups of criteria introduced into the general system, which have been defined according to the aspect of choosing a flat (house) variant, referring to the preferences of potential buyers and future users [1]. Figure 2 presents the overall block diagram of algorithms used by ESORD, using the variant ranking methods [5,6].

ESORD's task is to organize the possible variants, using the entire group of implemented algorithms of assessment methods or using only those indicated by the user (decision-maker). This approach has the advantage that the user receives a visual presentation of the results, a table and a classification of the considered variants (using each of the multi-criteria assessment methods) [5].

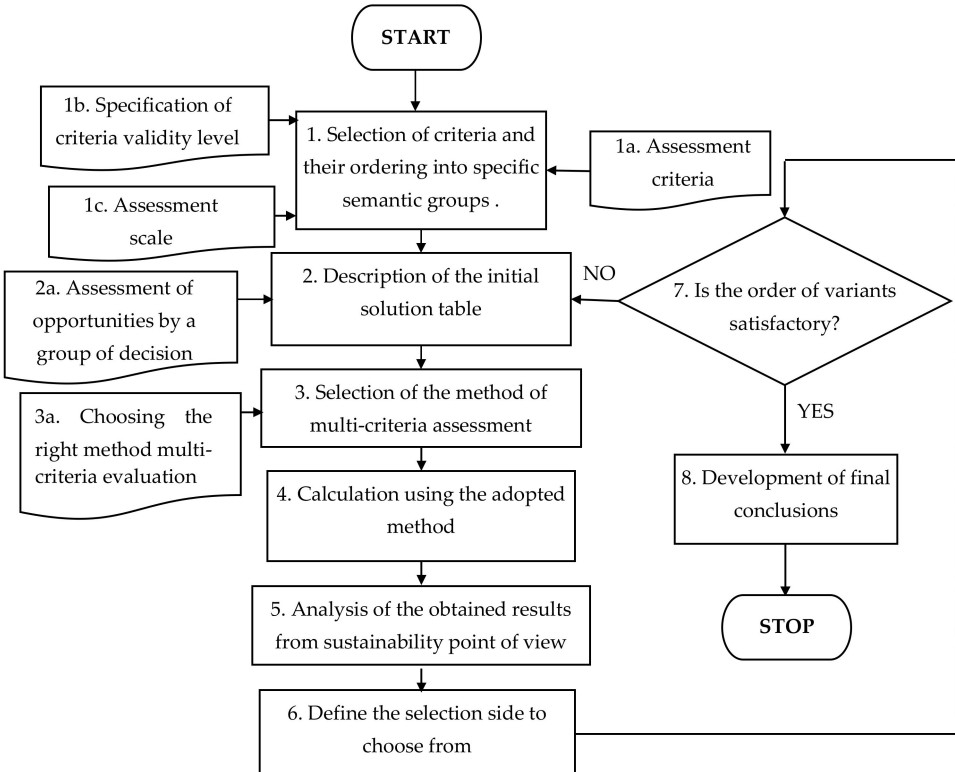

**Figure 2.** ESORD algorithm—assessment of decision variants.

In order to organize the pages of the project, multi-criteria assessment methods are used: AHP, ELECTRE, weighted sum, ideal point, and the fuzzy logic method. The ESORD software uses the above algorithms and the detailed results of the surveys in order to prioritize the examined criteria [1]. The computer algorithm calculates the main vector of weights for the examined criteria in order to prioritize their fulfillment. To achieve this type of vector of tested parameters (assessments) for the tested criteria: $K^L$—user assessment Type I for the assessment criterion; $K^{In}$—user assessment Type I for the criterion technical infrastructure of the facility; $K^{Ko}$—assessment of a Type I user in the category of facility structure criterion; $K^F$—assessment of Type I user for the rooms' functionality criterion; $K^S$—user Type I assessment for the criterion of the apartment finishing standard; $K^B$—Type I user security assessment; $K^C$—user assessment Type I in the cleanliness and ecology assessment category of the facility; $K^O$—Type I user assessment for attitude towards the object; $K^K$—user Type I assessment under cost category; $W^L$—main weights vector index value for the facility location criterion. The individual weighting reference points are calculated by the formula in Equation [5]:

$$W^{NK} = \frac{K^{NK}}{(K^L + K^{In} + K^{Ko} + K^F + K^S + K^B + K^C + K^O + K^K)} \tag{1}$$

where:

NK—mark (name) of a considered criterion

For example the value of $W_i^L$ index in relation to the facility location criterion is calculated using the following formula:

$$W^L = \frac{K^L}{(K^L + K^{In} + K^{Ko} + K^F + K^S + K^B + K^C + K^O + K^K)} \tag{2}$$

The system does not include the weights value for detailed criteria while calculating the main weights vector. The received results for the main criteria are presented in Table 1.

**Table 1.** The main criteria weights vector.

| Criterion | Weight | Criterion | Weight |
|---|---|---|---|
| Facility location | 0.143 | Safety | 0.143 |
| Technical infrastructure of the facility | 0.107 | Ecology and cleanness of the facility | 0.107 |
| Facility structure | 0.036 | Attitude towards the facility | 0.107 |
| Rooms' functionality | 0.143 | Costs | 0.107 |
| Apartment's finishing standard | 0.107 | | |

In fact, investors' needs (different investors with different attitudes to the risk and specific expectations) have to be taken under consideration and ESORD allows for it. The impact of weight criteria is described by Mashunin in [7,8]. The ESORD methodology has internally implemented calculation algorithms, selected methods of multi-criteria assessment, including methods of passing and weighted credit (sum and weighted sum), the ideal point method, the ELECTRE method, the fuzzy logic calculation method, and the AHP method. Each criterion is assessed in accordance with the following equation:

$$k_{ij}^{NK} = \left(O_{ij}^{I}\right)_{NK}^{2} \cdot \left(O_{ij}^{II}\right)_{NK}^{2} \tag{3}$$

where:

$k_{ij}^{NK}$—the value of the assessment itself for the NK criteria (list of indicators and its sub-criteria on the examples below).

$\left(O_{ij}^{I}\right)_{NK}$—assessment of NK assessment observing the guarded exemptions of the user Type I.

$\left(O_{ij}^{II}\right)_{NK}$—user assessment of NK Type II.

$n$—N unknown sub-criteria strings under a given NK criterion.

The introduction of the system of increasing the score of the variant under consideration to the power is necessary to enable the choice of differentiation of the score. In order to calculate the considered values, the decision possibilities for the ESORD criteria of the considered evaluation are calculated in accordance with the following purchases:

$$W_{w}^{NK} = \frac{\sum\limits_{n=1}^{r} \left(k_{ij}^{NK}\right)_{n}}{\sum\limits_{n=1}^{r} \left(\max k_{ij}^{NK}\right)_{n}}; \quad w = (1, \ldots, 10), \quad n = 1, 2, \ldots r \tag{4}$$

where:

$W_{w}^{NK}$—evaluation of the "w" variant according to the "NK" criteria

It is worth mentioning that during the creation of the algorithm system, the basic analysis for the assessment of residential buildings was separated and shaped (Table 2), along with the prioritization of the requirements of future tenants [6].

**Table 2.** The basic group of criteria used for assessment of residential construction facilities [1].

| A. Facility Location | B. Technical Infrastructure of the Facility | C. Facility Structure | D. Rooms' Functionality |
|---|---|---|---|
| A.1. City infrastructure (transport)<br>A.1.1. Subway (distance up to 1 km)<br>A.1.2. City train (distance up to 1 km)<br>A.1.3. Tramways (distance up to 1 km)<br>A.1.4. Buses distance up to 1 km)<br><br>A.2. Social infrastructure (service and education facilities)<br>A.2.1. Shopping and service facilities (food store, fair, market) up to 200 m<br>A.2.2. Education facilities (school, kindergarten, playground)<br>up to 200 m | B.1. Number of lifts in one stairway<br>B.1.1. Number of floors below 5 (except ground floor and underground) (1 lift)<br>B.1.2. –//– (2 lifts),<br>B.1.3. –//– (none)<br>B.1.4. Number of floors below 5 (except ground floor and underground) (1 lift),<br>B.1.5. –//– (2 lifts)<br>B.2. Car park<br>B.2.1. By the building,<br>B.2.2. Guarded car park<br>B.2.3. Park place inside the facility /multi-level car park,<br>B.2.4. Other place | C 1. Wall<br>C 2. Backbone<br>C 3. Mixed | D.1. Ergonomics of the rooms<br>D.1.1. Shape of rooms similar to square—easy to arrange<br>D.1.2. Niche for a wardrobe in the hall<br>D.1.3. Niche for a refrigerator in the kitchen<br>D.1.4. Other facilities (wardrobe, utility room, etc.)<br>D.1.5. Shape of rooms similar to a rectangle (long and narrow)<br>D.2. Height of the floors in residential premises<br>D.2.1. Up to 2.70 m<br>D.2.2. 2.71–2.90 m<br>D.2.3. Over 2.90 m |
| A3. Surrounding<br>A.3.1. Recreation areas (park, woods, lake, etc.) distance up to 1 km<br>A.3.2. Highway—distance up to 1 km<br>A.3.3. Recreation and sports centers—distance up to 1 km<br>A.3.4. Production plant (e.g., factory) with certain nuisance degree—distance up to 1 km | B.3. IT Telecommunication connections<br>B.3.1. Cable phone,<br>B.3.2. Cable TV,<br>B.3.3. Internet<br>B.3.4. Others | | |

| E. Apartment's Finishing Standard | F. Safety | G. Sustainability of the Facility | H. Attitude towards the Facility | I. Costs |
|---|---|---|---|---|
| E.1. Internal walls<br>E.1.1. Plaster + paint<br>E.1.2. Plaster E 1.3. other solution<br>E.2. Ceilings<br>E.2.1. Plaster + paint<br>E.2.2. Plaster<br>E.2.3. Other solution | F.1. Staircase<br>F.1.1. Entry phone<br>F.1.2. Videophone<br>F.2. Entry door to the apartment<br>F.2.1. Ordinary<br>F.2.2. Anti-burglary | G.1. Frequency of cleaning inside the facility<br>G.1.1. All working days<br>G.1.2. Three times a week<br>G.1.3. Two times a week<br>G.1.4. Other | H.1. Aesthetics of the facility exteriors<br>H.1.1. Plaster elevation<br>H.1.2. Glass elevation<br>H.1.3. Natural stone elevation<br>H.1.4. Other solution | I.1. Purchase price and utilization price<br>I.1.1. Sales price for 1 sq.m. of usable space of the building<br>I.1.2. The expected costs of utilization of the facility for 1 sq.m. of usable space of the building<br>I.1.3. Other costs |
| E.3. Floors<br>E.3.1. Made of plastics<br>E.3.2. Made of wood<br>E.3.3. Made of ceramic materials E.3.4. Made of stone materials<br>E.3.5. No finishing<br>E.4. Window sills<br>E.4.1. Made of plastics (for example conglomerate)<br>E.4.2. Made of stone materials (for example marble)<br>E.4.3. Metal<br>E.4.4. Wooden<br>E.4.5. Other | F.3. Building security<br>F.3.1. Receptionist inside the building<br>F.3.2. Guard at the entrance to the building<br>F.3.3. 24 h patrols of local security<br>F.3.4. Closed housing estate<br>F.3.5. Police precinct (distance—1 km) | G.2. Ecology<br>G.2.1. Waste selection<br>G.2.2. Electric energy saving devices in the facility (energy saving bulbs, movement sensors)<br>G.2.3. Water saving devices in the facility (meters)<br>G.2.4. Protection against noise<br>G.2.5. Water processing plant<br>G.2.6. Solar batteries<br>G.2.7. Other solutions | H.2. Comfortable use of the premises<br>H.2.1. Feeling of satisfaction and comfort<br>H.2.2. Feeling of safety<br>H.2.3. Other impressions | |

### 2.2. Monte Carlo Method

The use of simulation techniques makes it possible to increase the accuracy of the estimates of the effect of various scenarios on the project parameters. The most frequently used simulation is the Monte Carlo method, which enables the numerical simulation of various types of phenomena, in which there are input and output variables, but whose analytical description is complex. The essence of the Monte Carlo technique is the numerical determination of the cumulative distribution function for the value of the output variable using the probability density function assigned to the values of the input variables [9].

In simple terms, the Monte Carlo simulation technique consists of randomly selecting the values of the input variables, subject to the assumed probability distributions, and then calculating the value of the output variable. Repeated iteration of this procedure leads to the mapping of the probability distribution of the output variable. Monte Carlo simulation is an extremely useful and comprehensive form of probability analysis. It is based on the assumption that individual risk-related factors can be described by probability distributions. The name Monte Carlo refers to games based on repeated random attempts, such as roulette or dice [10].

The simulation model in the Monte Carlo method may be presented in the form of a spreadsheet, presenting, for example, an analysis of the duration of activities or a financial model of an investment project. This model is analyzed many times (usually 100 to 100,000 times). In each iteration (simulated process flow), the generated random number is assigned to the project parameters, in accordance with the assumed probability distribution. The results of the analysis are presented in the form of the resulting probability distribution. Using the Monte Carlo simulation technique, for each examined factor a histogram is obtained for various scenarios of materialization of the identified risk factors of the project [11].

### 2.3. Net Present Value

The most commonly used criteria for assessing the economic efficiency of projects, including construction projects, is net present value (NPV) [3,12–14]. NPV can be presented in several different ways and is calculated as the difference between the current value of the flow of inflows $(i_0, i_1, i_2, \ldots, i_T)$ and the current value of the expenditure stream $(e_0, e_1, e_2, \ldots, e_T)$ or net cash flow ($NCF_t$) for the period from the present moment ($t = 0$) to the planning horizon ($T$):

$$NPV = \sum_{t=0}^{T} \frac{i_t}{(1+r)^t} - \sum_{t=0}^{T} \frac{e_t}{(1+r)^t} = \sum_{t=0}^{T} \frac{i_t - e_t}{(1+r)^t} = \sum_{t=0}^{T} \frac{NCF_t}{(1+r)^t} = \sum_{t=0}^{T} NCF_t \delta^t \tag{5}$$

where $r$ is an interest rate, $\delta$ is a discount rate, and:

$$\frac{1}{(1+r)^t} = \delta^t. \tag{6}$$

The calculated NPV value can be interpreted, for example, as a surplus of net present revenues over the incurred initial expenditures or the investor's profit resulting from the implementation of the investment, taking into account changes in the value of money over time. NPV provides premises for making investment decisions. Most often a project is accepted if its $NPV > 0$ and rejected if $NPV < 0$. Projects with $NPV = 0$ are treated as neutral. It is important to mention information asymmetries, which strongly influence the value of NPV [15].

### 2.4. Original Method

The original method consists of several steps. At first, a basic cash-flow schedule is created for the construction project. It has a form of a spreadsheet with supporting documents. Then several variants of construction projects are tested and compared with the use of the ESORD IT tool to establish their

scores. These assessments affect the expected probability with which potential customers are willing to buy/rent a flat or commercial premises in the analyzed facility. The spreadsheet is updated with additional variants, their scores, and time and cost data. Probability distributions are assigned to the sales forecast.

The decision variable in the resulting model is the number (ID) of the selected project variant. The assumptions subject to a probability distribution are rental of the office and residential space (sales). Having developed a computational model, Monte Carlo simulations are performed to determine the probability distribution of cash flows and NPV of the project implementation.

Several approaches can be used to select the final project variant. For example, selecting the project with the highest expected NPV (aggressive investor) or selecting the project that has the greatest chance of obtaining a positive NPV (conservative investor) [16]. In Chapter 3 a case study is presented to show the effectiveness of the method.

## 3. Results

The investor has a possibility to construct a residential building. Ten different localizations and variants (in accordance with the spatial development plan) are taken into consideration. The considered variants are presented in Table 4. The elaborated data is available in [1]. The calculations were made in an Excel spreadsheet with the use of OptQuest®Engine, OptTek Systems, Inc.

Taking into account the ESORD calculation methodology, the IT system generated the results of selecting specific investment options, subject to balanced demand assessments. The method also allows for portfolio optimization, when an investor would like to get involved in more than one investment [17]. Table 3 provides a visual representation of the evaluation of variants generated by computer algorithms for a specific evaluation methodology.

**Table 3.** The collective presentation of variant assessments for all methods and final score.

| Variant | Weighted Average | Average | Ideal Point | Electre | Fuzzy Logic | AHP | Final Score |
|---------|------------------|---------|-------------|---------|-------------|-----|-------------|
| 1 | 0.114 | 0.115 | 0.123 | 0.172 | 0.120 | 0.121 | 518 |
| 2 | 0.107 | 0.106 | 0.116 | 0.207 | 0.112 | 0.104 | 409 |
| 3 | 0.091 | 0.093 | 0.092 | 0.0001 | 0.077 | 0.090 | 10 |
| 4 | 0.097 | 0.097 | 0.105 | 0.0001 | 0.104 | 0.095 | 71 |
| 5 | 0.097 | 0.098 | 0.074 | 0.138 | 0.084 | 0.096 | 67 |
| 6 | 0.102 | 0.100 | 0.099 | 0.034 | 0.096 | 0.102 | 120 |
| 7 | 0.105 | 0.104 | 0.115 | 0.103 | 0.118 | 0.101 | 286 |
| 8 | 0.102 | 0.101 | 0.102 | 0.103 | 0.113 | 0.104 | 194 |
| 9 | 0.083 | 0.084 | 0.063 | 0.0001 | 0.057 | 0.084 | 0 |
| 10 | 0.103 | 0.103 | 0.110 | 0.241 | 0.121 | 0.102 | 320 |

Final scores obtained by the ESORD software were later used to make predictions of sales. The better the score, the higher probability of customers renting a flat or office space.

On the basis of the collected data, cash-flow schedules were created. Sample NPV calculations are presented in Figure 3. In the presented case, NPV was calculated for the period of 15 years and the final score is presented in the right-lower corner of the figure. The discount rate used for the example calculation was 4%.

**Table 4.** Analyzed variants of the investment.

| **Variants** |
| --- |

Name: Variant 1    City: Warsaw
Area: Boremlowska 51
Average living area (m$^2$): 46
Average rent per m$^2$: 49 EUR/month
No. of flats: 190    Cost: 16.5 M EUR
Sample visualizations:

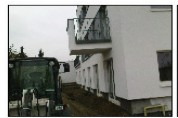 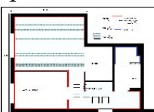 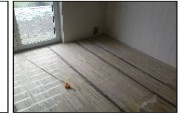
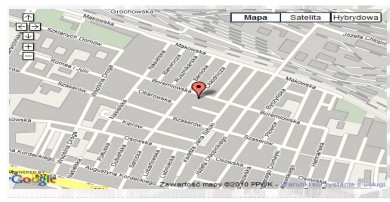

Name: Variant 2    City: Warsaw
Area: Kłopotowskiego 7
Average living area (m$^2$): 78
Average rent per m$^2$: 46 EUR/month
No. of flats: 150    Cost: 20 M EUR
Sample visualizations:

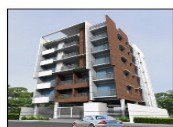 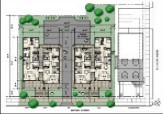 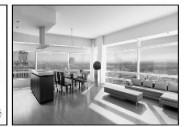
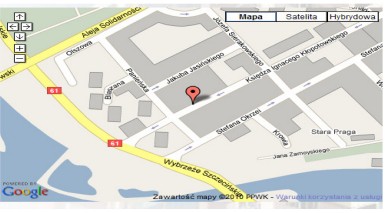

Name: Variant 3    City: Warsaw
Area: Grochowska 309
Average living area (m$^2$): 84
Average rent per m$^2$: 39 EUR/month
No. of flats: 680    Cost: 75 M EUR
Sample visualizations:

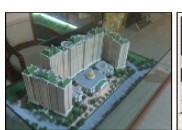 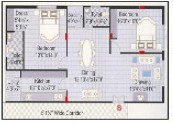 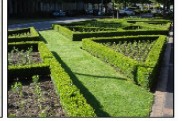
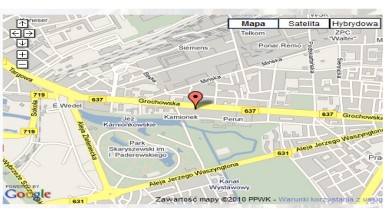

Name: Variant 4    City: Warsaw
Area: Bernardyńska 8
Average living area (m$^2$): 56
Average rent per m$^2$: 41 EUR/month
No. of flats: 150    Cost: 9 M EUR
Sample visualizations:

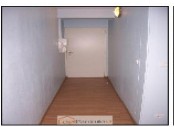 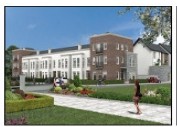 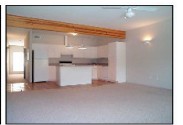
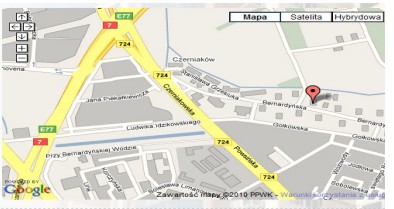

Name: Variant 5    City: Warsaw
Area: Broniewskiego 68
Average living area (m$^2$): 55
Average rent per m$^2$: 40 EUR/month
No. of flats: 210    Cost: 12.5 M EUR
Sample visualizations:

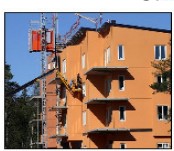 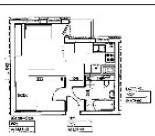 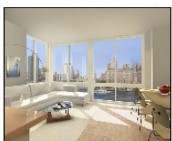
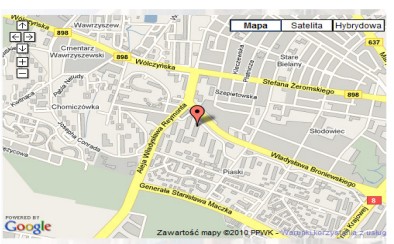

**Table 4.** *Cont.*

| Variants |
| --- |

Name: Variant 6     City: Warsaw
Area: Radiowa 16
Average living area (m$^2$): 76
Average rent per m$^2$: 37 EUR/month
No. of flats: 450     Cost: 40 M EUR
Sample visualizations:

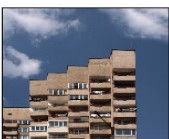 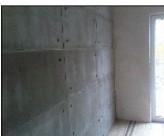 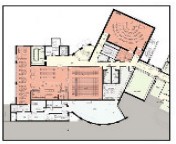 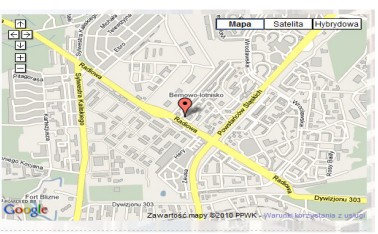

Name: Variant 7     City: Warsaw
Area: Polna 7
Average living area (m$^2$): 42
Average rent per m$^2$: 46 EUR/month
No. of flats: 150     Cost: 9 M EUR
Sample visualizations:

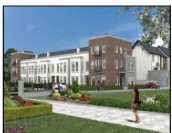 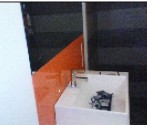 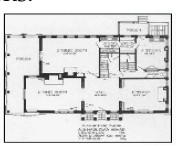 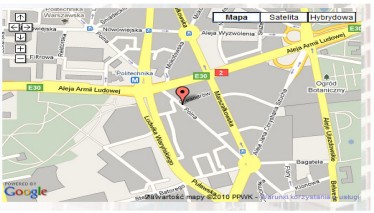

Name: Variant 8     City: Warsaw
Area: Alternatywy 4
Average living area (m$^2$): 69
Average rent per m$^2$: 42 EUR/month
No. of flats: 110     Cost: 9.625 M EUR
Sample visualizations:

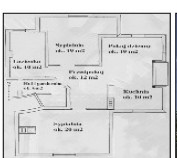 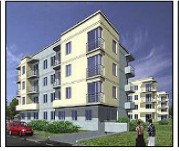 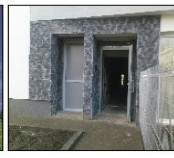 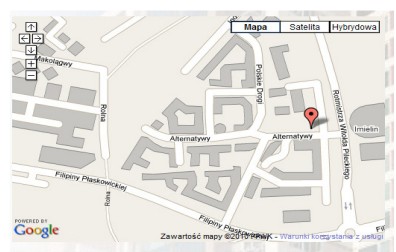

Name: Variant 9     City: Warsaw
Area: Wspólna 41
Average living area (m$^2$): 65
Average rent per m$^2$: 45 EUR/month
No. of flats: 160     Cost: 17.5 M EUR
Sample visualizations:

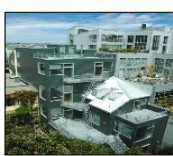 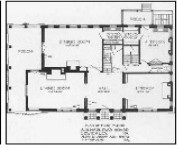 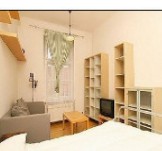 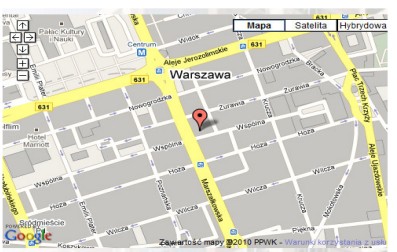

Name: Variant 10     City: Warsaw
Area: Zamieniecka 8
Average living area (m$^2$): 82
Average rent per m$^2$: 49 EUR/month
No. of flats: 110     Cost: 12 M EUR
Sample visualizations:

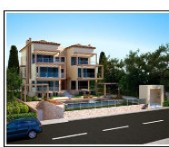 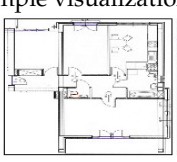 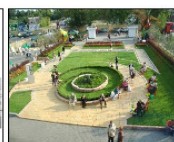 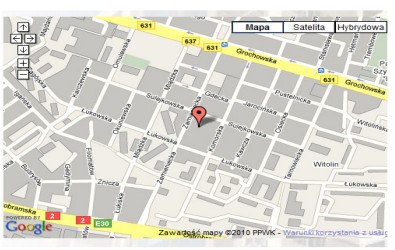

| | Year "0" | | 1 | | 2 | | ... | 14 | | 15 | |
|---|---|---|---|---|---|---|---|---|---|---|---|
| | 2021 | | 2022 | | 2023 | | ... | 2035 | | 2036 | |
| Sale (total lease of space) | € | - | € | - | € | 1 003 161.89 | ... | € | 2 911 060.19 | € | 2 911 060.19 |
| Rental of office space | | 0.00% | | 0.00% | | 38.75% | ... | | 88.75% | | 88.75% |
| Rental of residential space | | 0.00% | | 0.00% | | 28.75% | ... | | 88.75% | | 88.75% |
| One off bank provision | € | 144 375.00 | | | | | ... | | | | |
| Bank credit management | € | - | € | 212.86 | € | 212.86 | ... | € | 212.86 | € | 212.86 |
| Interest payment - margin 1,95% | € | 846.68 | € | 846.68 | € | 846.68 | ... | € | 846.68 | € | 846.68 |
| Main capital payment | € | 651 291.67 | € | 42 572.77 | € | 42 572.77 | ... | € | 42 572.77 | € | 42 572.77 |
| Premises construction | € | - | € | 9 625 000.00 | € | - | ... | | | | |
| Discounted premises price | € | 651 291.67 | € | 626 241.99 | € | 602 155.76 | ... | € | 376 104.71 | € | 361 639.14 |
| Land value minus depreciation | € | - | € | -14 403.57 | € | -28 253.15 | ... | € | -158 232.50 | € | -166 550.20 |
| Premises depreciation yearly | € | 14 979.71 | € | 14 403.57 | € | 13 849.58 | ... | € | 8 650.41 | € | 8 317.70 |
| Land tax | € | 543.79 | € | 543.79 | € | 543.79 | ... | € | 543.79 | € | 543.79 |
| Presmises tax | € | 13 967.35 | € | 13 967.35 | € | 13 967.35 | ... | € | 13 967.35 | € | 13 967.35 |
| Perpetual usefruct costs | € | - | € | 8 500.00 | € | 8 500.00 | ... | € | 8 500.00 | € | 8 500.00 |
| Ongoing maintenance costs | € | 24 000.00 | € | - | € | 85 268.76 | ... | € | 247 440.12 | € | 247 440.12 |
| Investments and capitalization of goodwill | € | - | € | - | € | 30 094.86 | ... | € | 87 331.81 | € | 87 331.81 |
| Net salaries per annum - 8 employees | € | 312 000.00 | € | 321 360.00 | € | 331 000.80 | ... | € | 383 720.65 | € | 383 720.65 |
| Cost of keeping the books of accounts | € | 6 240.00 | € | 6 427.20 | € | 6 620.02 | ... | € | 7 674.41 | € | 7 674.41 |
| Employee social wages per year - 8 employees | € | 156 000.00 | € | 160 680.00 | € | 165 500.40 | ... | € | 191 860.32 | € | 191 860.32 |
| Legal service | € | - | € | 30 000.00 | € | 30 000.00 | ... | € | 30 000.00 | € | 30 000.00 |
| others | € | 1 200.00 | € | - | € | 4 263.44 | ... | € | 12 372.01 | € | 12 372.01 |
| Income tax | € | - | € | -14 403.57 | € | 176 751.18 | ... | € | 544 451.03 | € | 544 783.74 |
| Cash flow | € | -1 310 464.48 | € | -10 195 707.07 | € | 107 019.00 | ... | € | 1 339 566.41 | € | 1 339 233.70 |
| Discount factor | | 1 | | 0.96 | | 0.92 | ... | | 0.58 | | 0.56 |
| Discounted cash flow | € | -492 406.54 | € | -9 803 564.49 | € | 98 945.08 | ... | € | 773 566.23 | € | 743 628.94 |
| Cumulated discounted cash flow | € | -492 406.54 | € | -10 295 971.03 | € | -10 197 025.95 | ... | € | 948 276.84 | € | 1 691 905.78 |
| NPV= | € | -492 406.54 | € | -10 295 971.03 | € | -10 197 025.95 | | € | 948 276.84 | € | 1 691 905.78 |

**Figure 3.** Sample cash flow and NPV calculations [own source].

ESORD scores were normalized (scale from −1 to 1) and Beta PERT distribution was used to modify predicted likeliness of customers renting premises. Sample distributions are presented in Figure 4.

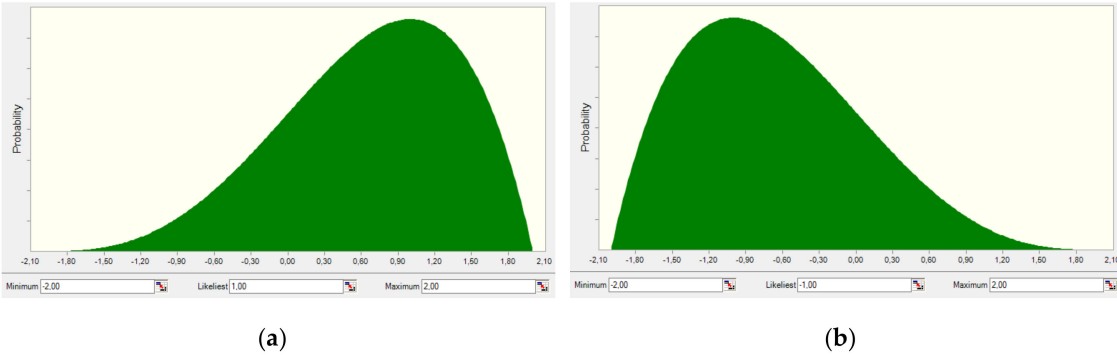

(**a**)            (**b**)

**Figure 4.** Sample distributions modifying predicted sales: (**a**) Variant 1—highest ESORD score; (**b**) Variant 9—lowest ESORD score [own source].

After completing the spreadsheet and modeling non-deterministic parameters, computational simulations were started. In the considered case, it was decided to optimize the financial schedule by maximizing the expected NPV. For comparison, the likelihood of obtaining a positive NPV was also maximized. The best results are presented in Figures 5 and 6. The forecast data percentiles comparison of the two best variants is presented in Table 5. Statistics of the results are compared in Table 6.

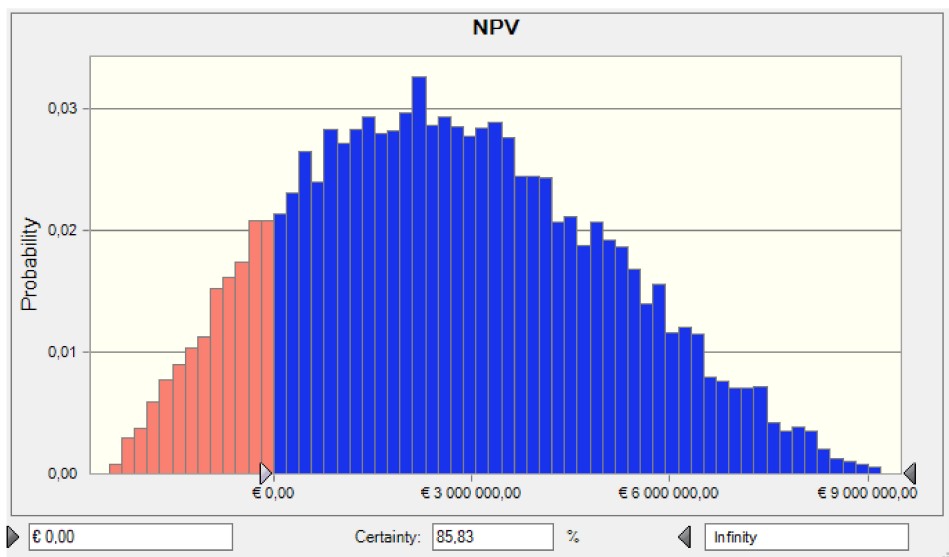

**Figure 5.** NPV forecast: maximizing the final value of net present value (NPV)—best variant—6. [own source].

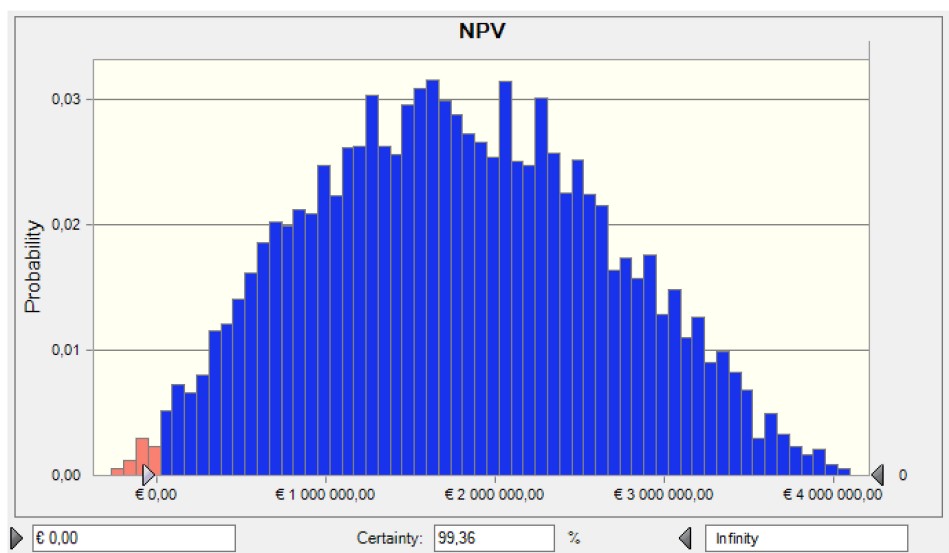

**Figure 6.** NPV forecast: maximizing the probability that NPV is greater than or equal to € 0.00—best variant—8.

**Table 5.** NPV forecast—percentiles comparison.

| Percentiles | Forecast Values—Variant 6 | Forecast Values—Variant 8 |
|---|---|---|
| 0% | € −2,485,290.30 | € −263,575.36 |
| 10% | € −384,060.74 | € 637,532.01 |
| 20% | € 474,991.99 | € 982,691.91 |
| 30% | € 1,205,920.74 | € 1,266,688.35 |
| 40% | € 1,882,102.82 | € 1,520,634.29 |
| 50% | € 2,514,830.58 | € 1,753,883.31 |
| 60% | € 3,185,442.91 | € 2,017,660.81 |
| 70% | € 3,889,059.27 | € 2,277,573.71 |
| 80% | € 4,766,356.07 | € 2,567,897.49 |
| 90% | € 5,852,536.29 | € 2,968,311.24 |
| 100% | € 9,840,775.14 | € 4,090,146.43 |

**Table 6.** NPV forecast statistics.

| Statistics | Forecast Values—Variant 6 | Forecast Values—Variant 8 |
| --- | --- | --- |
| Trials | 10,000 | 10,000 |
| Base case | € 580,141.89 | € 1,691,905.78 |
| Mean | € 2,638,797.81 | € 1,785,113.04 |
| Median | € 2,514,842.14 | € 1,754,597.81 |
| Standard deviation | € 2,341,475.23 | € 867,874.75 |
| Variance | € 5,482,506,257,658.74 | € 753,206,577,194.92 |
| Skewness | 0.2299 | 0.1155 |
| Kurtosis | 2.42 | 2.35 |
| Coeff. of variability | 0.8873 | 0.4862 |
| Minimum | € −2,485,290.30 | €−263,575.36 |
| Maximum | € 9,840,775.14 | € 4,090,146.43 |
| Range width | € 12,326,065.44 | € 4,353,721.79 |
| Mean std. error | € 23,414.75 | € 8,678.75 |

The last step of the procedure assumes the selection of the final project variant. The presented solutions correspond to the choices of the aggressive investor (variant 6) and the conservative investor (variant 8).

## 4. Discussion

The proposed approach integrates the utility and sustainability assessment tool (ESORD) with cash-flow schedules and the Monte Carlo method. Thanks to the developed approach, investors are able to optimize their decisions obtaining the best results. Importantly, the approach allows users to adjust the approach to the assumptions of the decision-maker, allowing you to make a decision that corresponds to the character of the investor (e.g., aggressive or conservative approach). Due to its stochastic nature, the procedure can be particularly useful in uncertain times, allowing traders to analyze the whole range of possible events. Future research in the discussed field can be related to other important elements of decision making [18]: for example, how to incorporate the investor's risk acceptance level, more detailed characteristics of investors, etc.

**Author Contributions:** Conceptualization, M.K.-N. and J.R.; methodology, P.N.; software, M.K.-N., J.Z., P.N. and J.R.; validation, M.K.-N.; formal analysis, P.N.; investigation, M.K.-N., J.Z., P.N. and J.R.; resources, M.K.-N., J.Z., P.N. and J.R.; data curation, M.K.-N., J.Z., P.N. and J.R.; writing—original draft preparation, M.K.-N., J.Z., P.N. and J.R.; writing—review and editing, M.K.-N., J.Z., P.N. and J.R.; visualization, J.R.; supervision, M.K.-N. All authors have read and agreed to the published version of the manuscript.

**Funding:** This research received no external funding.

**Conflicts of Interest:** The authors declare no conflict of interest.

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
