# Peer review of "Cash-Flow Schedules Optimization within Life Cycle Costing (LCC)"

_sustainability, doi:10.3390/su12198201_

Round 1

Reviewer 1 Report

Review for the authors of Article 9

Authors investigate a problem of " Title: Cash Flow Schedules Optimization within LCCs".

This paper proposes a decision-making model for the implementation of the Investment and construction plan, architectural and construction solutions, spatial and technological solutions that are used in the early stages of the project. According to the authors, such a model has a significant impact on the achievement of the investment goals of investors who optimize their decisions and customer expectations.

The authors for order to organize the pages of the project, multi-criteria assessment methods are used: AHP, ELECTRE, weighted sum, ideal point and the fuzzy logic method.

The authors use the ESORD program, which evaluates detailed results of the surveys in order to prioritize the examined criteria.

The result is a set of weighting factors, which are presented in Table 1. The main criteria weights vector.

But the authors do not understand the influence of "weighting factors" on the result of the decision (as well as the authors of literary references). The authors suggest that the weighting factors equalize the criteria for making decisions. The more weight, the more important (priority) the corresponding criterion. For example, from table 1

  1. Ecology and cleanness of the facility KC =0,107 «a»
  2. Facility structure KKo =0,036 «b»

it follows that KC =0,107 is three times more important than KKo =0,036.

Let's analyse the result mathematically.

The value of the first indicator indicates "a" and the value of the second indicator indicates "b." The work assumes that the weights equalize their effect on the task. Then the relation is carried out

KC *a = KKo *b

or

0.107*a=0.036*b

From here

b=0.107*a/0.036 or

b=3*a.

As a result, we learn that the second criterion is 3 times more effective than the first criterion, i.e. the result with accuracy vice versa.

In work

https://rdcu.be/bhZ8i in the application

I conducted a theoretical analysis of the use of weights and showed that they give quite the wrong result that the author puts into them.

Therefore, all further arithmetic actions will lead to the answer that the authors need.

Thus, in general, the review of the article is negative.

Recommendations.

At the same time, I believe that the article should be printed in your journal (together with the review).

I will explain.

For the first time I published the theory and methods of vector optimization in the book:

"Methods and Models of Vector Optimization." - M.: Science, 1986. 141 p., that is, more than thirty years ago. All these thirty years I have proved to everyone the correctness of solving the vector problem. And only in work

https://rdcu.be/bhZ8i

they recognized its theoretical significance.

But before publishing the work, I analysed over a hundred papers. And found out their shortcomings (all of them did not solve the problem as a whole).

Based on the analysis, an axiom was formulated:

"Equality and equivalence of criteria in a vector (multi-criteria) problem of mathematical programming". The axiom took five lines, but it made it possible to construct the optimality principle and solve the vector problem.

If it were not for these hundred (not correct) works, I would never have formulated my axiom in five lines.

The volume of work shows that the authors worked on the article. In science (and mathematics especially), a negative result is also a result. It is quite possible that I did not see something that will interest readers.

Therefore, I recommend the work to be published together with the review.

See also a monograph in English:

Mashunin Yu. K. Theory and Methods of Vector Optimization. (Volume One). Cambridge Scholars Publishing. London, 2020, p. 183.

Mashunin Yu.K.

Author Response

Dear Prof. Mashunin,

Many thanks for constructive comments and detailed mathematical analysis. In fact investors needs (differen investors with differen attitude to the risk) has to be taken under consideration and our system allows for it. Article can be publish with the review, we fully agree.

Best Regards

Authors

Reviewer 2 Report

This manuscript covers an interesting topic and its written form is superb. I would have several comments which could improve the quality of this manuscript. I am very optimistic about the publishable opportunity of this manuscript. 

1. Although the authors claimed that the information (variants of the investment) would influence the NPV, I also would suggest that the authors should include a caveat regarding the existence of information asymmetries, which probably influences the value.

  • Huynh, T. L. D., Wu, J., & Duong, A. T. (2020). Information Asymmetry and firm value: Is Vietnam different?. The Journal of Economic Asymmetries21, e00147.

2. The author should explicitly indicate the interest rate, which was used to calculate the NPV. 

3. The authors should expand your discussion in conclusion part. To be more precise, how do the authors learn from your findings and results? I would suggest that the authors could draw the future pictures of this field research. For example, how to incorporate the risk appetite, the characteristics of investors, etc. Your work would have some merits to introduce the potential outlets for the researchers

  • Huynh, T. L. D. (2020). Replication: Cheating, loss aversion, and moral attitudes in Vietnam. Journal of Economic Psychology, 102277.

4. Figure 2 (Sample cash flow and NPV calculation) would be informative in the good quality rather than using the picture (I think the picture quality is not good). 

5. In Introduction, the authors claimed "This aspect of the solution is especially important in uncertain times like during COVID-19 pandemic." It holds true when the authors assert some extant literature how your study could solve the COVID-19 puzzle.

  • Huynh, T. L. D. (2020). Does culture matter social distancing under the COVID-19 pandemic?. Safety Science, 104872.

6. I have no objection to the back-testing of NPV with normal distribution. However, the authors would benefit to have the portfolio optimization in terms of the investors who want to buy more than one investment items. 

  • Nguyen, S. P., & Huynh, T. L. D. (2019). Portfolio optimization from a Copulas-GJRGARCH-EVT-CVAR model: Empirical evidence from ASEAN stock indexes. Quantitative Finance and Economics3(3), 562.

7. I found some typographical errors in your papers, please revise it carefully (especially - line 59). 

All in all, let me conclude that I enjoyed reading your manuscript and I think the revised version would be promising to resonate among the academia network. 

Author Response

Dear Reviewer No 2

Many thanks – all reviewer 2 comments were taken under consideration. English improved by native speaker. Some minor revisions were done. ALL CHANGES ADVISED BY REVIEWER 2 – marked in BLUE in the newly uploaded article.

Reviewer 3 Report

The following contents are to be added or improved:

The article has some minor linguistic errors.
Definition of the Cash Flow Schedule.
It would be important to submit some information about the existing systems of Cash Flow Schedules Optimization with reference to the corresponding literature sources.
Description of the ESORD-IT-System.

Author Response

Dear Reviewer No 3

We agree with the third rewiever. Linguistic minor improvements were done. Definition of the Cash Flow Schedule was written. Some references related to Cash Flow Schedules Optimization were researched and added. Description of the ESORD-IT-System was written.

ALL CHANGES DONE, as ADVISED BY REVIEWER 3 – marked in yellow in the newly uploaded article.

Best Regards

Authors